# Current Concepts in Endoscopic Bladder Neck Injection: Combined Antegrade and Retrograde Endoscopic Injection of the Bladder Neck in Children with Neurogenic Bladder

**DOI:** 10.3390/children9040449

**Published:** 2022-03-23

**Authors:** Frank-Martin Haecker, Anja Mettler, Alexander Mack

**Affiliations:** 1Department of Pediatric Surgery, Children’s Hospital of Eastern Switzerland, 9006 St. Gallen, Switzerland; anja.mettler@kispisg.ch (A.M.); alexander.mack@kispisg.ch (A.M.); 2Faculty of Medicine, University of Basel, 4001 Basel, Switzerland

**Keywords:** urinary incontinence, neurogenic bladder, antegrade and retrograde endoscopic bladder neck injection

## Abstract

Introduction: Urinary incontinence is common in patients with neurogenic bladder, and efficient management is an ongoing challenge. Besides open surgical procedures like bladder neck reconstruction, artificial sphincter implantation, or sling procedures, endoscopic bladder neck injections of bulking agents enable minimally invasive access with promising results. Several studies report on the effect of antegrade vs. retrograde endoscopic injection techniques. We report our preliminary experience of combined antegrade and retrograde endoscopic injection of the bladder neck in children with neurogenic bladder, in selected cases combined with intravesical Botox^®^ injection. Methods: With the patient in lithotomy position, antegrade urethrocystoscopy was performed using a 9.5 Fr cystoscope. In parallel, percutaneous suprapubic bladder access introducing a second 9.5 Fr. cystoscope was accomplished. Four quadrant Dx/H injections were performed, with the two surgeons guiding each other by parallel endoscopy to the optimal localization for injection. In selected patients, the procedure was completed with transurethral intravesical Botox^®^ injection. Results: A total of 6 children underwent the combined procedure (2/6 patients including intravesical Botox^®^ injection). The mean follow-up was 15 months (range 3 to 48). 5 Patients experienced a significant improvement of urinary incontinence, however one patient demonstrated complete failure. Conclusions: Even if we present only preliminary results with a limited number of patients, we present a minimally invasive technique with encouraging results. In carefully selected patients, combined antegrade and retrograde endoscopic injection of the bladder neck is a useful tool to treat urinary incontinence.

## 1. Introduction

Permanent urinary incontinence is common in patients with neurogenic bladder due to congenital disorders such as meningomyelocele (MMC). MMC patients may suffer mild to moderate symptoms only from stress incontinence. However, many MMC patients complain about permanent urinary incontinence, independent from bladder volume, body posture/positioning, and activity. Efficient management of the incompetent bladder neck presents an ongoing challenge to Pediatric urologists. There are different methods to treat urinary incontinence, all of which are still discussed controversially. During the last few decades, open surgical repair such as bladder neck reconstruction, artificial sphincter implantation, or sling procedures represent the preferred methods of treatment. In 1985, Vorstman et al. reported their preliminary experience using endoscopic Polytetrafluorethylene injection for urinary incontinence in children [1]. Within the last 30 years, endoscopic treatment of vesico-ureteral reflux (VUR) has become an established alternative to long-term antibiotic prophylaxis and even to open ureteral reimplantation in selected patients. Different tissue augmenting substances such as polydimethylsiloxan, polytetrafluoroethylene, autologous fat, and collagen have been utilized. Since Stenberg reported his first experience with dextranomer gel/hyaluronic acid (Dx/HA) in 1995 [2], multiple studies using Dx/HA for endoscopic treatment of VUR have been published. Subsequently, Dx/HA was also used for endoscopic bladder neck injection [3,4,5,6,7,8]. In contrast to open repair, endoscopic bladder neck injections of bulking agents enable minimally invasive access with promising results. The majority of these studies reported either on antegrade or on retrograde bladder neck injection. Dean et al. compared the effect of antegrade vs. retrograde bladder neck injection [7], and a recent published study confirmed similar results with a success rate of dry patients as 35% [9]. The proportion of significantly improved patients tended to be higher after the antegrade rather than the retrograde bladder neck injection technique [9]. This experience is in contrast to DaJusta et al., who reported a success rate of only 25% [10]. However, these patients underwent bladder neck injection after a failed sling procedure [10].

We report our preliminary experience of combined antegrade and retrograde endoscopic injection of the bladder neck in children with permanent urinary incontinence. In selected patients, the procedure was combined with intravesical Botox^®^ injection.

## 2. Methods

Preoperative evaluation included the patient’s history, clinical examination, urine culture, renal ultrasound, incontinence charts, and videourodynamics, which were repeated during routine follow-up visits, with the schedule depending on the patient’s individual clinical course. None of the patients underwent any previous surgical bladder neck procedure. Indication for combined antegrade and retrograde endoscopic bladder neck injection was based on persistent urinary incontinence despite intensive conservative therapy including Oxybutynin medication (orally or intravesical) and/or after intravesical Botox^®^ injection. In patients presenting with inadequate bladder capacity and/or decreased compliance, combined antegrade and retrograde endoscopic injection was completed with transurethral intravesical Botox^®^ injection (10 IU/kg BW, max. 300 IU).

With the patient under general anesthesia in the lithotomy position, antegrade urethrocystoscopy was performed using a 9.5 Fr cystoscope (Figure 1). In parallel, percutaneous suprapubic bladder access introducing a second 9.5 Fr. cystoscope was accomplished. Four quadrant Dx/HA injections were performed, with the two surgeons guiding each other by parallel endoscopy to the optimal localization for injection (Figure 2, Figure 3 and Figure 4). In selected patients, the procedure was completed with transurethral intravesical Botox^®^ injection. At the end of the procedure, a transurethral Foley catheter was placed and left in place for 10 days. Patients were discharged on the first day after surgery and they returned to the outpatient clinic for scheduled removal of the Foley catheter.

## 3. Results

A total of 6 patients (5 girls, 1 boy, aged 9 to 14 years) underwent combined antegrade and retrograde bladder neck injection. Neurogenic bladder with incompetent bladder neck, deficiency of low pressure storage function, and/or low compliance bladder was the underlying condition in 5 patients, as well as bladder exstrophy with a wide open incompetent bladder neck in 1 patient. MMC patients used clean intermittent catherization (CIC) for controlled and complete daily bladder emptying. The male patient with exstrophy micturated by detrusor controlled contraction.

Transurethral cystoscopy displayed a wide open bladder neck at the beginning of the procedure (Figure 5). Under guidance by antegrade suprapubic endoscopy, transurethral retrograde bladder neck injection was performed at the 3 o’clock and 9 o’clock positions (Figure 6). Subsequently, bladder neck injection at the 6 o’clock and 12 o’clock positions was performed through antegrade percutaneous suprapubic endoscopy. Direct antegrade and retrograde visualization confirmed effectiveness of the luminal occlusion (Figure 7). As a (positive) consequence of effective injection, visibility of the optimal location for injection was compromised with increasing volume of the bulking agent. The combined injection technique is helpful to reduce this limitation as much as possible.

Fortunately, we did not observe any postoperative complication like wound infection, wound dehiscense, bleeding, hematuria, etc., in this small study population.

The mean follow-up was 15 months (range 3 to 48). Two patients achieved complete dryness, and another 2 patients experienced a significant improvement of urinary incontinence. One patient showed temporary dryness for 3 months afterwards, which was still an improvement in comparison to preoperative condition for another 6 months. One patient demonstrated complete failure after removal of the Foley catheter. These two patients underwent a second combined bladder neck injection procedure, with one patient demonstrating significant improvement during another 9 months follow-up and one patient demonstrating complete failure. 

## 4. Discussion

Efficient management of the incompetent bladder neck with consecutive permanent urinary incontinence presents still an ongoing challenge to Pediatric urologists. For decades, open surgical repair such as bladder neck reconstruction, artificial sphincter implantation, or sling procedures were the preferred methods of treatment. With the widespread use of endoscopic injection of bulking agents to correct VUR, this treatment modality became more popular and the technique was extended to bladder neck injection. Endoscopic injection treatment at the bladder neck is less invasive than open surgical procedures, well tolerated, and relatively easy to perform. The safety and effectiveness of the ideal bulking agent could be confirmed by multiple studies correcting VUR. The majority of studies/authors reporting about endoscopic injection of the bladder neck used Dx/HA, which is also preferred in our institution.

In the current literature, the majority of studies are retrospective studies reporting on a small number of patients (maximum 51 patients [9]) including an inhomogeneous study population (extrophy/epispadia, neurogenic bladder with/without catherizable urinary stoma, etc.). Some authors report on gender specific different success rates. Different bulking agents were also used. Dyer et al. found no difference between the effectiveness of Teflon vs. Dx/HA [6]. In their experience, endoscopic bladder neck injection was ineffective and expensive [6]. In contrast, Misseri et al. found dryness in approx. 20% of their study population and significant improvement in another 30% [3]. However, follow-up was short at 9.5 months. We could confirm their results with similar success rates with a longer follow-up.

We consider the combination of the antegrade and retrograde injection technique as a useful option to treat patients with permanent urinary incontinence. Even if the efficacy is limited, we consider this technique as an option for all patients who are poor surgical candidates and those who want to avoid extensive bladder neck reconstruction. 

## 5. Conclusions

Even if we present only preliminary results with a limited number of patients, we present a minimally invasive technique with encouraging results. In carefully selected patients, combined antegrade and retrograde endoscopic injection of the bladder neck is a useful tool to treat urinary incontinence.

## Figures and Tables

**Figure 1 children-09-00449-f001:**
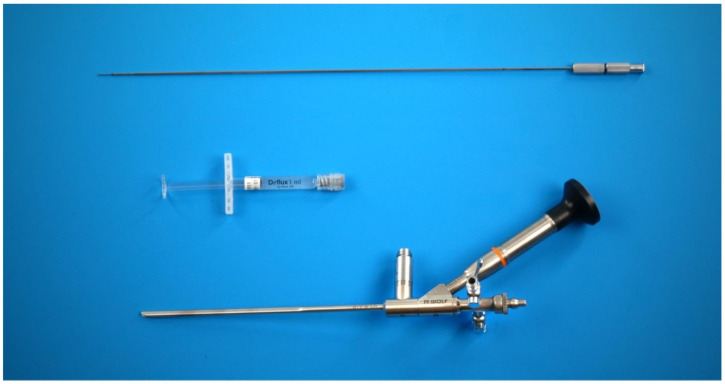
9.5 Fr cystoscope with metal needle (2 sets needed).

**Figure 2 children-09-00449-f002:**
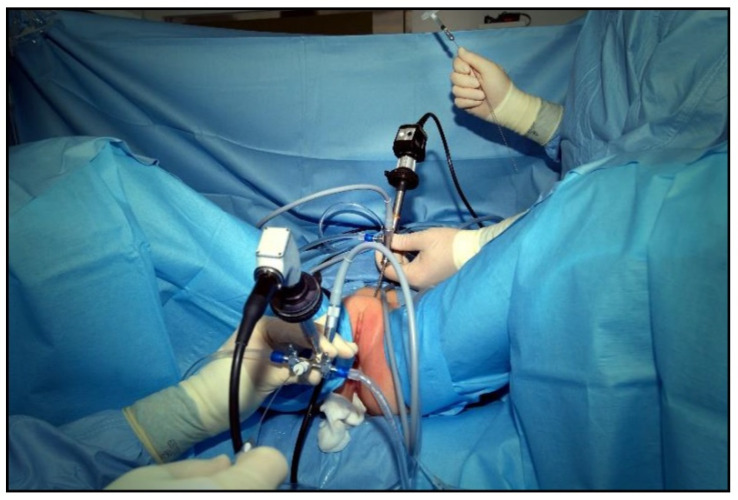
Combined antegrade and retrograde endoscopy.

**Figure 3 children-09-00449-f003:**
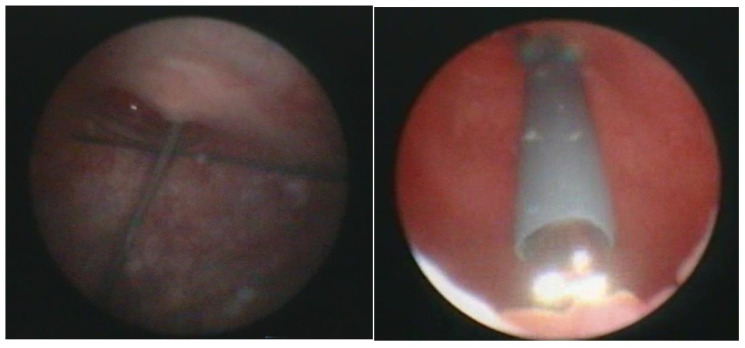
Transurethral cystoscopy controlling the percutaneous suprapubic bladder access.

**Figure 4 children-09-00449-f004:**
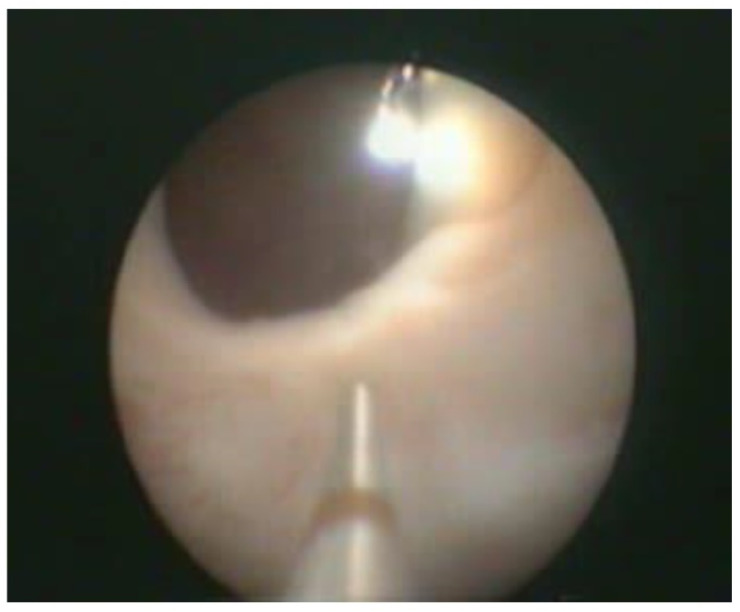
Combined endoscopy to guide each other for optimal localization for bladder neck injection.

**Figure 5 children-09-00449-f005:**
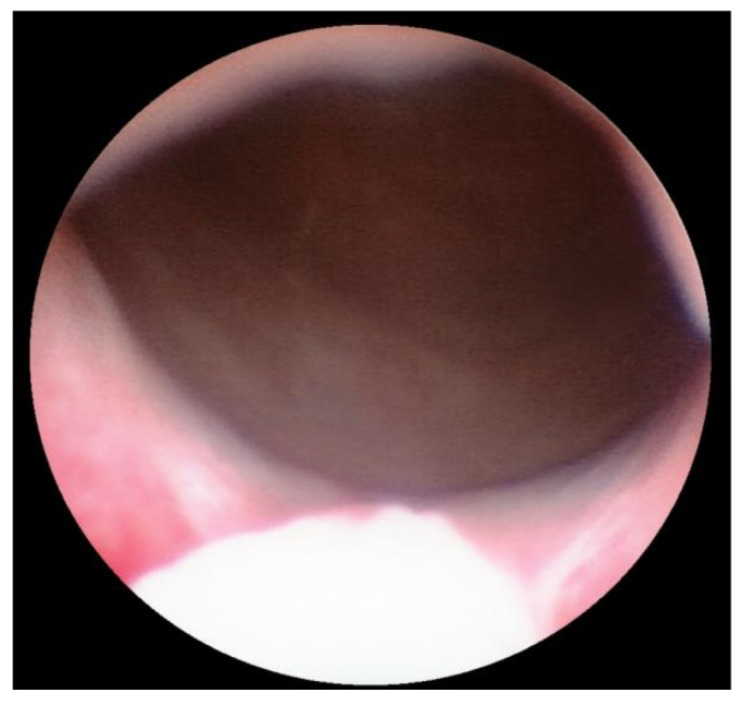
Transurethral cystoscopy: wide open bladder neck.

**Figure 6 children-09-00449-f006:**
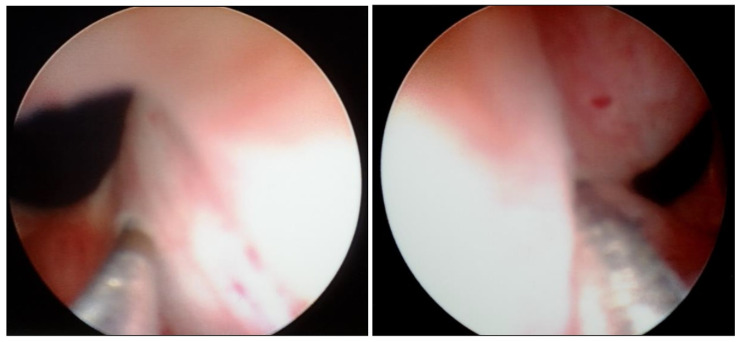
Transurethral retrograde bladder neck injection at 3 o’clock and 9 o’clock position.

**Figure 7 children-09-00449-f007:**
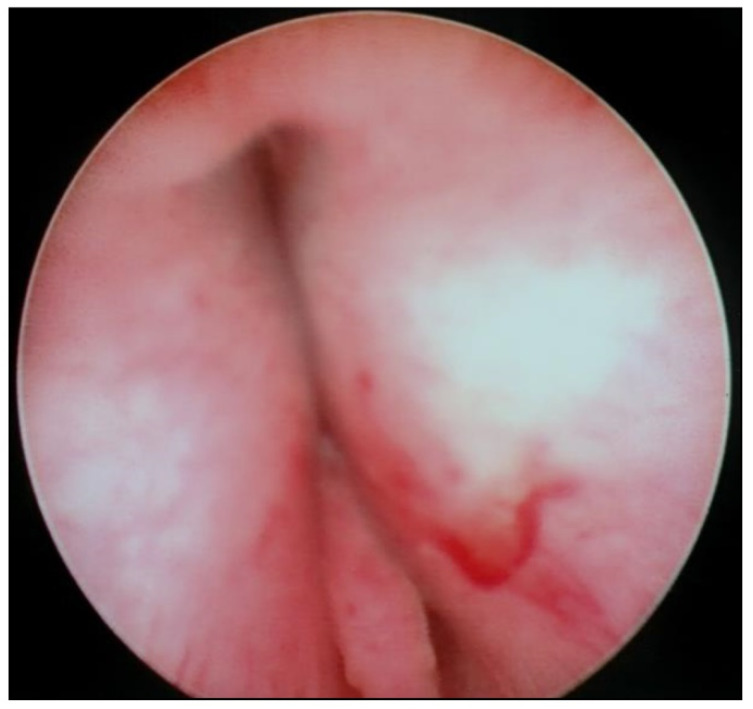
Retrograde cystoscopy confirming effectiveness of luminal occlusion (before injection: see Figure 5).

## Data Availability

Not applicable.

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
