# Peer review of "Current Concepts in Endoscopic Bladder Neck Injection: Combined Antegrade and Retrograde Endoscopic Injection of the Bladder Neck in Children with Neurogenic Bladder"

_children, 2022, doi:10.3390/children9040449_

Round 1
Reviewer 1 Report
The reviewer wants to be informed of the causal disease of the neurogenic bladder in the five patients. The information about the methods of urine drainage in the six patients is also required.
Basically, urinary incontinence in neurogenic bladder is attributed to lower urinary tract disfunction of two categories, that include the incompetency of the internal and/or external urethral sphincter mechanism and the deficiency of low pressure storage function of urinary bladder as a result of detrusor overactivity and/or low compliance bladder. Which pathology were the patients suffered from ? Was there any relationship between the pathology of incontinence and the effectiveness of the treatment ?
The reviewer would like to have more detailed explanation why and how the guidance by antegrade and retrograde endoscopy contributed to the optimal performance of the injection from the other direction.
Author Response
Dear Prof. Bergholz, dear Estelle Ding
Thank you again for your messages including the valuable reviewer comments for our manuscript “Combined antegrade and retrograde endoscopic injection of the bladder neck in children with neurogenic bladder” Children-1626534.
We revised the manuscript and added the information as requested. Please find our point-by-point reply below.
Reviewer 1:
The reviewer wants to be informed of the causal disease of the neurogenic bladder in the five patients. The information about the methods of urine drainage in the six patients is also required.
Basically, urinary incontinence in neurogenic bladder is attributed to lower urinary tract disfunction of two categories, that include the incompetency of the internal and/or external urethral sphincter mechanism and the deficiency of low pressure storage function of urinary bladder as a result of detrusor overactivity and/or low compliance bladder. Which pathology were the patients suffered from ? Was there any relationship between the pathology of incontinence and the effectiveness of the treatment ?
The reviewer would like to have more detailed explanation why and how the guidance by antegrade and retrograde endoscopy contributed to the optimal performance of the injection from the other direction.
Details about the origin of neurogenic bladder and the type of urinary incontinence as well as an explanation about the advantage of the combined antegrade and retrograde injection technique were added.

Reviewer 2 Report
Thank you for considering me as a reviewer for this publication. I have provided my comments as follows.
General Comments:
I think this article is substantial because it describes a very interesting topic about a rare type of endoscopic treatment in children.
Working with children who have urinary incontinence and other LUTS is very difficult, especially in children with neurological diseases.
The authors investigated the use of dextranomer gel/hyaluronic acid in the endoscopic injection therapy in children with neurogenic urinary incontinence, which contributes to existing knowledge, and is especially important because very little has been written about it in the scientific literature.
The paper was written based on a sample of 6 children which is small, but the authors have explained that this is just a preliminary experience of „a minimally invasive technique with encouraging results“ and that „in the current literature, the majority of studies are retrospective studies reporting on a small number of patients including inhomogeneous study population“.
This article can be accepted for publication but some spelling mistakes need to be corrected (see the Specific comments).
This article will be useful for pediatric surgeons, pediatricians, urologists, nephrologists, and other physicians that treat voiding disorders in children.
Specific Comments:
Abstract
Line 12. man-agement
Line 17. neuro-genic
Line 20. cys-toscope
Line 22. pro-cedure
Line 28. ante-grade
1. Introduction
The authors have not described what type of urinary incontinence it was. Considering that they mentioned the incompetent bladder neck and that they described the therapy with bulking agents and Botox, it could be concluded that it was isolated stress incontinence and 2 cases of mixed urinary incontinence. I think that the authors should define the term „neurogenic bladder“ and describe the etiology and the symptoms of urinary incontinence in children with neurological disease.
Methods (the title number is missing) 2. Methods
Results (the title number is missing) 3. Results
It should be stated whether the patients were only girls or boys, or both sexes. The age of the children should also be stated.
Here the authors again describe the evaluation procedures and operational techniques that should be written in the chapter on Methods.
I think it should be stated here which neurological diseases caused the neurogenic bladder in children.
It might be good to mention if there were complications like infections, bleeding, etc.
Line 87. ... at the begin of the = ... at the beginning of the procedure
Discussion (the title number is missing) 4. Discussion
Conclusion (the title number is missing) 5. Conclusion
References
It is necessary to edit the references because they are not uniform.

Author Response
Dear Prof. Bergholz, dear Estelle Ding
Thank you again for your messages including the valuable reviewer comments for our manuscript “Combined antegrade and retrograde endoscopic injection of the bladder neck in children with neurogenic bladder” Children-1626534.
We revised the manuscript and added the information as requested. Please find our point-by-point reply below.
Reviewer 2:
Abstract
Line 12. man-agement
Line 17. neuro-genic
Line 20. cys-toscope
Line 22. pro-cedure
Line 28. ante-grade
- Introduction
The authors have not described what type of urinary incontinence it was. Considering that they mentioned the incompetent bladder neck and that they described the therapy with bulking agents and Botox, it could be concluded that it was isolated stress incontinence and 2 cases of mixed urinary incontinence. I think that the authors should define the term „neurogenic bladder“ and describe the etiology and the symptoms of urinary incontinence in children with neurological disease.
Methods (the title number is missing) 2. Methods
Results (the title number is missing) 3. Results
It should be stated whether the patients were only girls or boys, or both sexes. The age of the children should also be stated.
Here the authors again describe the evaluation procedures and operational techniques that should be written in the chapter on Methods.
I think it should be stated here which neurological diseases caused the neurogenic bladder in children.
It might be good to mention if there were complications like infections, bleeding, etc.
Line 87. ... at the begin of the = ... at the beginning of the procedure
Discussion (the title number is missing) 4. Discussion
Conclusion (the title number is missing) 5. Conclusion
References
It is necessary to edit the references because they are not uniform.
Editorial revisions (s. Abstract line 12, line 17, etc.) were completed.
@Children team: no hyphens were in the original word-file. Could you please check the file after transfer to the template? Thank you.
Introduction and Results were revised/completed (see also above “reviewer 1). Since the aim of our manuscript is the description of a technical innovation as a current concept, we didn’t explain all details of symptoms of urinary incontinence. We hope that our provided additional information is sufficient.
References: we noticed some minor editorial mutations, probably due to the same technical problem as mentioned above. We revised the list of references.
